# Molecular Modeling and Potential Ca²⁺ Channel Blocker Activity of Diphenylmethoxypiperidine Derivatives

**Victor M. Pulgar** [1,2,3,*] , **Jill Harp** [2,4,5,6] **and Tony E. Reeves** [7]

1. Department of Pharmaceutical & Clinical Sciences, College of Pharmacy & Health Sciences, Campbell University, Buies Creek, NC 27506, USA
2. Biomedical Research Infrastructure Center, Winston-Salem State University, Winston-Salem, NC 27101, USA
3. Department of Obstetrics and Gynecology, Wake Forest School of Medicine, Winston-Salem, NC 27157, USA
4. Department of Biological Sciences, Winston-Salem State University, Winston-Salem, NC 27101, USA
5. Department of Chemistry, Winston-Salem State University, Winston-Salem, NC 27101, USA
6. Department of Physiology and Pharmacology, Wake Forest School of Medicine, Winston-Salem, NC 27157, USA
7. Department of Internal Medicine, Wake Forest School of Medicine, Winston-Salem, NC 27157, USA
* Correspondence: pulgar@campbell.edu; Tel.: +1-910-893-1701

**Abstract:** Molecular interactions of 4-diphenylmethoxy-1-methylpiperidine derivatives with the calcium channel CaV1.1 (pdb:6JP5) are described. All the compounds tested, previously shown to inhibit adrenergic vascular contractions, display similar binding energetics and interactions with the trans-membrane domain of 6JP5 on the opposite side relative to the channel pore, where nifedipine, a known dihydropyridine Ca²⁺ channel blocker binds. Additionally, the compounds tested inhibit Ca²⁺-dependent contractions in isolated mouse mesenteric arteries. Thus, diphenylpyraline analogs may exert their anticontractile effects, at least partially, by blocking vascular Ca²⁺ channels.

**Keywords:** piperidinol-4; L-type calcium channels; arterial contraction; diphenylmethoxypiperidine





## 1. Introduction

Voltage-dependent calcium (Ca²⁺) channels (VDCC) are expressed within a variety of excitable cells and regulate multiple physiological functions. In the cardiovascular system, VDCC play important roles in controlling vascular tone, enzymatic activities, and neurotransmitter release [1]. Since arterial tone is regulated by membrane potential, VDCC have a predominant role in regulating excitation–contraction coupling, thus vascular smooth muscle cells (VSMC) express, among others, the L-Type of VDCC (LTCC) that upon membrane depolarization display large-conductance, and long-lasting inward Ca²⁺ currents [2].

Ca²⁺ influx through LTCC is recognized as the main mediator of myogenic tone in VSMC. LTCC contains the pore-forming alpha 1c subunit (α1c) and auxiliary subunits β, α2δ, and γ that modulate channel function, and several studies have shown the importance of α1c in controlling vascular tone [1]. For example, selective inhibition of α1c by dihydropyridine antagonists abolishes increases in intracellular Ca²⁺ concentration ($[Ca^{2+}]_i$) and myogenic tone [2], and several of those antagonists have a relevant role in the treatment of cardiovascular diseases such as hypertension and angina pectoris [3]. Within the several LTCC isoforms present in the cardiovascular tissues, most of the antihypertensive and anti-ischemic effects of LTCC blockers are the result of inhibition of the Cav1.2 isoform, promoting decreases in peripheral vascular resistance and inotropy [4]. In addition to their cardiovascular effects, altered function of LTCCs has been linked to a range of neurologic diseases such as autism spectrum disorders and Parkinson's disease among others [5], highlighting the relevance of LTCCs as drugs targets.

We have previously shown that several 4-diphenylmethoxy-1-methylpiperidine derivatives share the ability to inhibit receptor independent and receptor dependent contractions in isolated arteries [6]. Thus, depolarization and adrenergic agonist-dependent contractions are attenuated in the presence of these compounds. Since VSMC contraction depends chiefly on influx of $Ca^{2+}$ ions, we tested the hypothesis that 4-diphenylmethoxy-1-methylpiperidine derivatives exert their anticontractile actions, at least in part, by interacting and inhibiting the activity of LTCC in small resistance arteries.

## 2. Materials and Methods

We have previously described the synthesis of these compounds [6]. 4-Diphenylmethoxy-1-methylpiperidine analogs, **2a**, **2b** and **4a** were used in these analyses (see Table 1). For the modelling studies, we used $Ca_V1.1$, the structure where the basis for $Ca^{2+}$ selectivity and the interactions of LTCC inhibitors dihydropiridines (DHPs) and phenylalkylamines with the channel have been defined [7,8]. The cryo EM structure for $Ca_V1.1$ (from *Oryctolagus cuniculus*, Rabbit) with nifedipine was obtained from the Research Collaboratory for Structural Bioinformatics (RCSB) Protein Data Bank [9]; PDB ID code: 6JP5 [10,11]. Structures were prepared and minimized following described procedures [12] for AutoDock Tools [13]. All co-crystallized molecules, ions, and water molecules were removed. Hydrogen atoms were added, and the protonation states on polar residues were optimized prior to protein minimization [14]. Ligands were prepared by adding hydrogens at pH 7.3 and 3D coordinates generated using Open Babel and docked to the 6JP5 structure. The docking of the 4-diphenylmethoxy-1-methylpiperidine derivatives was performed in an unbiased manner. The complete protein was included in the docking grid(s) to allow the ligands to sample the entire protein surface and water accessible cavities. The docking scores presented in the manuscript are for the site having the lowest free energy of binding. Sites with low affinity binding (above -4 kcal/mol) were not evaluated. All structural figures were prepared with PyMol [15]. Sequence alignments were performed using CLUSTAL [16].

**Table 1.** Binding energy of the interactions between of 4-diphenylmethoxy-1-methylpiperidine derivatives, nifedipine and $Ca_V1.1$. Structures of compounds **2a**, **2b**, and **4a** are shown on top of the table. Values for FW and predicted binding energy (kcal/mol) are listed.

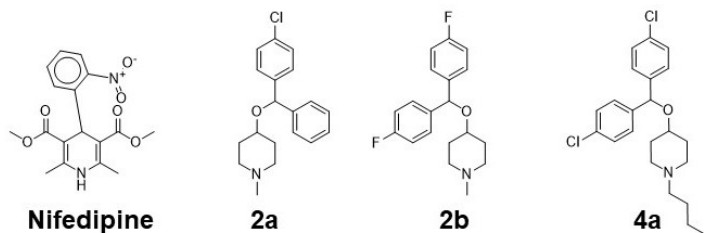

| Name | SMILES | FW g/mol | Kcal/mol |
|---|---|---|---|
| Nifedipine | COC(=O)C1=C(C)NC(=C(C1c1ccccc1[N+](=O)[O-])C(=O)OC)C | 346.3 | −5.8 |
| **2a** | CN3CCC(OC(c1ccccc1)c2ccc(Cl)cc2)CC3 | 315.8 | −8.2 |
| **2b** | CN3CCC(OC(c1ccc(F)cc1)c2ccc(F)cc2)CC3 | 317.4 | −8.3 |
| **4a** | CCCCN3CCC(OC(c1ccc(Cl)cc1)c2ccc(Cl)cc2)CC3 | 392.4 | −8.4 |

For vascular studies, $Ca^{2+}$-dependent isometric contraction was tested in isolated mouse mesenteric resistance arteries mounted in a multi wire myograph, as previously described [17]. Arterial segments were normalized, to calculate optimal diameter (OD), to $0.9 \cdot L_{100}$, with $L_{100}$ being the internal circumference the vessels would have if they were exposed to a transmural pressure of 100 mm Hg [18]. Arterial segments used had an OD = 290 ± 11 μm. After resting, maximal contraction with Krebs–Henseleit Buffer (KHB) containing 75 mM KCl, where all $Na^+$ was replaced by $K^+$. Arterial segments used display an average $K_{MAX} = 2.7 \pm 0.2$ mN/mm. After washing and resting, a $Ca^{2+}$-dependent concentration dose response curve was performed. Segments in basal

tone were exposed to KHB without $Ca^{2+}$ (KHB-Ca) for 20 min with one washing step at 10min with KHB-Ca, the 4-aryl methoxypiperidinol derivatives were added and incubation continued for 10 min. The buffer solution was changed to KHB-Ca containing 75 mM $K^+$, compounds were replenished, and incubation extended for an additional 15 min. After incubation, contraction was elicited by adding $Ca^{2+}$ in increasing steps from 0.03 to 2.78 mM with 2 min of contact time between doses. Concentration–response curves to $Ca^{2+}$ were analyzed by fitting individual experimental data to a logistic curve to determine maximal response and sensitivity. The curve was of the form $Y = \text{bottom} + (\text{top} - \text{bottom}) (1 + 10(\text{LogEC}_{50} - X) * \text{Hill Slope}))$ where X is the logarithm of the concentration of $Ca^{2+}$ and Y is the contractile response; maximal values were expressed as relative to the contraction in KHB in the presence of 75 mM KCl. Sensitivity was expressed as $pD_2$ ($pD_2 = -\log [EC_{50}]$) with $EC_{50}$ being the concentration of agonist producing 50% of the maximal response. One segment per animal was used to test each compound ($n = 7$). All protocols for animal handling were approved by Campbell University. Data are expressed as mean $\pm$ SEM. One-way analysis of variance (ANOVA) with Bonferroni's multiple comparisons was used to determine significant differences. A $p < 0.05$ was accepted as an indication of statistical significance. Optimal diameters (OD) were calculated as $OD = 0.9 \cdot L_{100} / \pi$. One arterial segment per animal was used to test each compound ($n = 7$). All protocols for animal handling were approved by Campbell University. Data are expressed as mean $\pm$ SEM. One-way analysis of variance (ANOVA) with Bonferroni's multiple comparisons was used to determine significant differences. A $p < 0.05$ was accepted as an indication of statistical significance.

## 3. Results

### 3.1. Molecular Interactions with Cav1.1

Our modelling studies predicted that the 4-diphenylmethoxy-1-methylpiperidine derivatives tested interact with the $Ca^{2+}$ channel structure used for the analyses. Binding energy values were higher than those calculated for the known VDCC inhibitor nifedipine (Table 1) with all three derivatives **2a**, **2b** and **4a** displaying similar energy of interaction ($-8.2, -8.3, -8.4$ kcal/mol, respectively)

### 3.2. Docking Analysis

Analysis of the docking results revealed that the 4-diphenylmethoxy-1-methylpiperidine derivatives were predicted to bind in a pocket adjacent to the nifedipine binding site in the transmembrane region of the 6JP5 structure (Figure 1A). Cluster analysis of the ligand poses in the highest affinity site identified specific residues for the interactions between 4-diphenylmethoxy-1-methylpiperidine derivatives and Cav1.1. As shown in Figure 1B, the piperidine methyl of each ligand was bound in a hydrophobic pocket comprising PHE 218, LEU 217 and ILE 305. Each ligand had a phenyl halide consistently coordinated with GLU 156. This binding pocket is significantly deeper and less solvent exposed than that of nifedipine, presumably contributing to the more favorable calculated binding energies, but by incurring in more structural rigidity in the protein and being more distal from the channel pore may account for the reduced activity compared to nifedipine.

### 3.3. Protein Sequence Alignment of LTCC Alpha Subunits

The Cav1.1 isoforms of the alpha subunit of the VDCC are highly conserved proteins displaying more than 90% sequence identity [19]. Sequence alignment between Cav1.1 isoforms from Homo sapiens (Human), Oryctolagus cuniculus (Rabbit), Rattus norvegicus (Rat) and Mus musculus (Mouse) showed that all residues predicted as important for binding to the 4-diphenylmethoxy-1-methylpiperidine derivatives tested are fully conserved between the Cav1.1 isoforms analyzed (Figure 2A). When comparing with other isoforms of the VDCC alpha subunit, sequence identity between Cav1.1 and Cav1.2 decreases at 66% [19]. Alignment between the Cav1.1 isoform from Oryctolagus cuniculus (Rabbit), the same isoform used for the docking studies, and the Cav1.2 from the same species, showed

that majority of the residues predicted are conserved. Exceptions include the substitutions ILE205MET and ALA1233THR, with both residues replaced by groups sharing strong similar properties, and TRP309LEU, showing the replacement by a different group (Figure 2B).

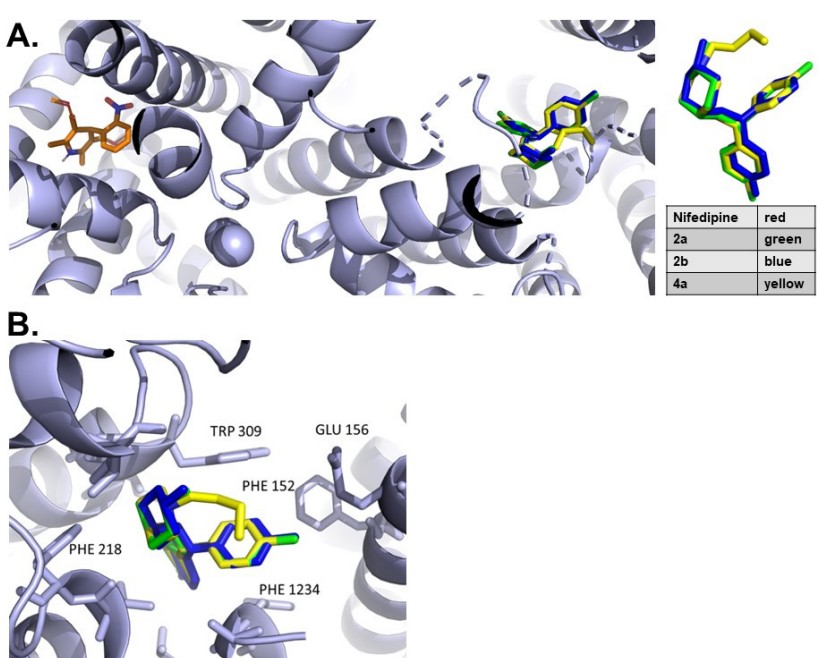

**Figure 1.** Docking results of 4-diphenylmethoxy-1-methylpiperidine derivatives, nifedipine with Cav1.1. (**A**). Each of the analogs (**2a**-green, **2b**-blue, **4a**-yellow) are predicted to bind in a more deeply buried pocket opposite in the calcium channel from the known inhibitor nifedipine (red). (**B**). Analysis of interactions of 4-diphenylmethoxy-1-methylpiperidine derivatives with Cav1.1. LEU 1237, PHE 1234, and ALA 1233 form a hydrophobic pocket that binds one of the phenyl groups and ILE 312, TRP 309 and PHE 152 form a pocket for the second. PHE 218, ILE 305 and LEU 217 comprise a hydrophobic pocket that interacts with the piperidine alkyl moiety of each ligand.

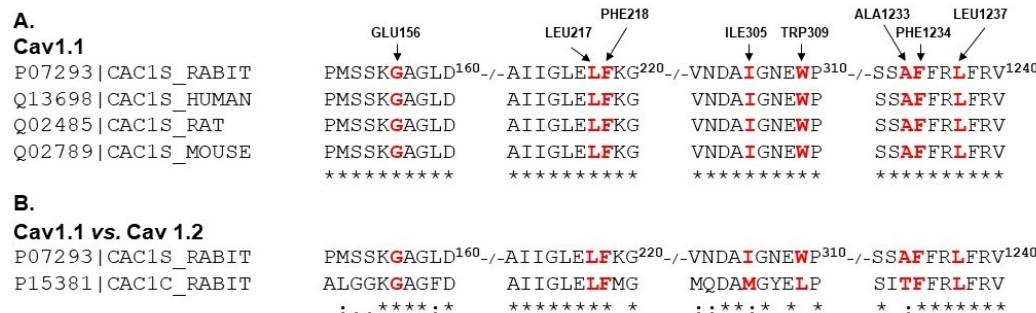

**Figure 2.** Protein Sequence alignment of VDCC alpha subunits. Protein sequences were retrieved from the UniProt KB database and aligned with CLUSTAL using default settings [16]. Sequence ID and names are shown on the left, sections of the alignment containing the residues predicted to contribute to the binding of 4-diphenylmethoxy-1-methylpiperidine derivatives (in red font) are shown. (**A**). Residues are indicated on top, numbering is following CAC1S_RABIT sequence and sequences aligned include Cav1.1 proteins from Oryctolagus cuniculus (Rabbit), Homo sapiens (Human), Rattus norvegicus (Rat) and Mus musculus (Mouse). (**B**). Alignment between Cav1.1 and Cav1.2 from Oryctolagus cuniculus (Rabbit). * = fully conserved residue, : = groups with strongly similar properties, . = groups with weakly similar properties.

### 3.4. Diphenylmethoxypiperidine Derivatives Block $Ca^{2+}$-Dependent Contractions

Our functional studies allowed us to test the $Ca^{2+}$-dependent contraction in isolated mouse mesenteric arteries. All the 4-diphenylmethoxy-1-methylpiperidine derivatives tested

inhibit $Ca^{2+}$-dependent contraction with variable efficacy (Figure 3, Table 2). Compared to the control, **2a**, **2b** and **4a** diminished maximal response ($E_{MAX}$) and sensitivity ($EC_{50}$) of $Ca^{2+}$-dependent contractions. As expected, preincubation with nifedipine (100 mM) completely blocked $Ca^{2+}$-dependent contractions. Since the concentration of nifedipine used was ten times lower (100 nM vs. 1 µM), the 4-diphenylmethoxy-1-methylpiperidine derivatives tested were significantly less effective at inhibiting contraction.

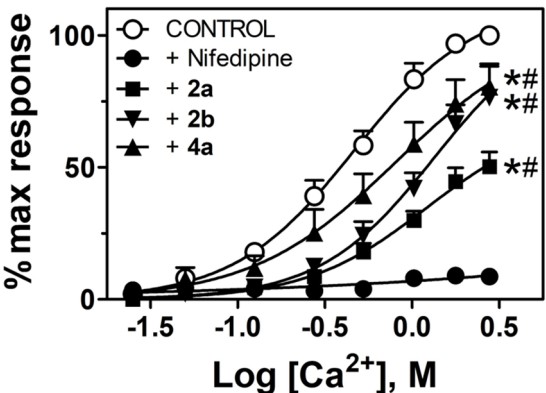

**Figure 3.** Effects of 4-diphenylmethoxy-1-methylpiperidine derivatives and nifedipine on $Ca^{2+}$-dependent contractions in mouse mesenteric arteries. Arterial segments ($n = 7$) in the presence of KHB without $Ca^{2+}$ and containing 75 mM KCl (KHB-Ca+K) were exposed to increasing concentrations of $Ca^{2+}$ (0.03–2.78 mM) until maximal response was reached. Parallel experiments included pre-incubation with **2a** (■), **2b** (▼) or **4a** (▲) all at 1 µM, or nifedipine (●, 100 nM). [$Ca^{2+}$] is expressed as Log[$Ca^{2+}$]. * $p < 0.05$ vs. Control in maximal response; # $p < 0.05$ in sensitivity.

**Table 2.** Maximal $Ca^{2+}$-dependent contraction and sensitivity in the presence of 4-diphenylmethoxy-1-methylpiperidine derivatives. Vascular responses in control mesenteric arteries and arteries pre-incubated with 4-diphenylmethoxy-1-methylpiperidine derivatives were measured as described. Maximal contraction ($E_{MAX}$) was expressed as %$K_{MAX}$ and sensitivity as $EC_{50}$ (mM). * $p < 0.05$ vs. Control.

|  | EMAX (%KMAX) | EC50 (mM) |
|---|---|---|
| Control | $113 \pm 2$ | $0.48 \pm 0.07$ |
| **2a** | $62 \pm 9$ * | $1.02 \pm 0.11$ * |
| **2b** | $87 \pm 7$ * | $1.2 \pm 0.07$ * |
| **4a** | $94 \pm 2$ * | $0.81 \pm 0.05$ * |

## 4. Discussion

Our study showed that 4-diphenylmethoxy-1-methylpiperidine derivatives inhibit $Ca^{2+}$-mediated ex vivo vascular contractions. To provide insights into potential binding interactions of the evaluated compounds with relevant mediators of $Ca^{2+}$-mediated contractility, we performed modeling analyses with the alpha subunit of LTCC. Our results predicted that 4-diphenylmethoxy-1-methylpiperidine derivatives interact with the $Ca^{2+}$ channel structure Cav1.1 in the opposite side of the pore compared to one of the clinically relevant LTCC blockers nifedipine. Docking studies showed that nifedipine binds to the side of Cav1.1 structure consistently with previous work indicating that this blocker binds to an allosteric site in the channel differently from the other families of blockers, such as phenylalkylamines, that bind closer to that channel's pore [10]. The docking studies also highlight important molecular features for potential optimization of small molecule-channel interactions. For instance, lengthening one of the phenyl halides into the pocket formed by LEU 1237 and GLU 156, would afford the opportunity to optimize additional electrostatic interactions.

The residues in Cav1.1 predicted to bind the 4-diphenylmethoxy-1-methylpiperidine derivatives are identical between four species compared, suggesting that this interaction is highly conserved. The possibility for the compounds analyzed to inhibit Cav1.2 isoforms is partially supported by the conservation of the residues predicted to be involved in their binding. Although the sequence identity is lower between Cav1.1 and Cav1.2 than within the Cav1.1 family [19], only one of the predicted amino acids is changed between Cav1.1 and Cav1.2 (TRP309LEU), suggesting that 4-diphenylmethoxy-1-methylpiperidine derivatives may also interact with and inhibit Cav1.2. This may expand the potential functions of these compounds beyond the vasculature given the described different tissue distributions of the isoforms Cav1.1 and Cav1.2, skeletal muscle versus smooth muscle, brain, heart, and adrenals, respectively [19]. Certainly, more studies are needed to ascertain the impact of the potential interactions of 4-diphenylmethoxy-1-methylpiperidine derivatives with the VDCC isoform Cav1.2.

We should keep in mind that the method used did not allow for a rank of the docking poses using binding energies predicted for the 4-diphenylmethoxy-1-methylpiperidine derivatives studied. Additional analyses using molecular mechanics with generalized Born and surface area solvation (MM/GBSA) would allow a re-rank of the docking poses and contribute to the design of compounds with better activity playing a relevant role in structure–activity relationship analyses [20].

It needs to be noted that differences in pKa between nifedipine (weak acid) and the 4-diphenylmethoxy-1-methylpiperidine derivatives (weak bases) suggest that at the physiological pH tested the percentage of ionization will differ considerably between them. This issue could affect the energetics of predicted drug-protein binding and it should be addressed in a larger library analysis; it certainly represents a limitation of the current study.

The role of VDCC and other $Ca^{2+}$ channels in health and disease is expanding and consequently the therapeutic potential of their modulation [21]. Since additional $Ca^{2+}$ channels such as T-type have been shown to play a role in modulating vascular tone [22], we cannot rule out that the 4-diphenylmethoxy-1-methylpiperidine derivatives tested could mediate their effects trough interaction with T-type $Ca^{2+}$ channels. The design of $Ca^{2+}$ channels blockers with better affinity and specificity would expand the possibilities to modulate the activity of these important channels. Our studies suggest structural changes to the diphenylpiperidine molecules, such as modifications to the N-alkyl moiety and/or substitutions on the aryl rings, may strengthen interactions and may optimize Cav1.1, and potentially Cav1.2 $Ca^{2+}$ channel inhibition and efficacy. Future structure–activity relationship studies will address these hypotheses.

## 5. Conclusions

Molecular interactions with the calcium channel subunit Cav1.1 may explain the ability of 4-diphenylmethoxy-1-methylpiperidine derivatives to partially inhibit $Ca^{2+}$-dependent contractions. This activity can contribute to the inhibition of adrenergic-dependent vascular contraction exhibited by these compounds.

**Author Contributions:** Conceptualization V.M.P.; methodology V.M.P. and T.E.R.; investigation, V.M.P. and T.E.R.; data curation, V.M.P., J.H. and T.E.R.; writing—original draft preparation, V.M.P.; writing—review and editing, V.M.P., J.H. and T.E.R. All authors have read and agreed to the published version of the manuscript.

**Funding:** This research was funded by the Department of Pharmaceutical & Clinical Sciences, Campbell University and MD00232 from the National Institutes of Health. The APC was funded by the Biomedical Research Infrastructure Center (BRIC) Winston-Salem State University.

**Data Availability Statement:** Docking results can be made available through Figshare under tereeves@wakehealth.edu.

**Acknowledgments:** The facilities of the Winston Salem State University Biomedical Research Infrastructure Center (BRIC) are greatly appreciated.

**Conflicts of Interest:** The authors declare no conflict of interest.

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
