# Peer review of "Molecular Modeling and Potential Ca2+ Channel Blocker Activity of Diphenylmethoxypiperidine Derivatives"

_chemistry, doi:10.3390/chemistry5020050_

Round 1

Reviewer 1 Report

The manuscript submitted by Pulgar et al., investigates the potential Ca2+ channel blocker activity of diphenylmethoxypiperidine derivatives. The authors used molecular docking to characterize the molecular interactions of three diphenylmethoxypiperidine derivatives with a LTCC channel  and a functional study to investigate their influence on the Ca2+-dependent contractions in the mouse mesenteric arteries. After a careful study of the manuscript, I found some concerns in the study and therefore, I regret to inform you that I am unable to consider the manuscript for publication in this journal.

Please see the following comments:

1) One of the serious concerns of this study is that the diphenylmethoxypiperidine derivatives  don't show as effectiveness as nifedipine in functional studies and not even by 50% effective compared to nifedipine. This shows that these compounds are not effective to block the LTCC channels despite binding to one of the binding regions of LTCC channel with high affinity. The manuscript clearly missed the details of mechanism of action of how diphenylmethoxypiperidine derivatives binding in that region can trigger the blockage of the channel.

2)    The following statement from lines 143-146 is incorrect and misleading as diphenylmethoxypiperidine derivatives don't interact at the same position as nifedipine and don’t block the channel as effectively  as nifedipine.

“Our results showed that 4-diphenylmethoxy-1-methylpiperidine derivatives interact

with Ca2+ channel structure Cav1.1 in a manner similar to one of the clinically relevant LTCC blockers nifedipine.”

3)     How did the authors choose the binding site? Why did the authors consider the binding pocket adjacent to nifedipine for the 4-diphenylmethoxy-1-methylpiperidine derivatives and these details are missing. How did the authors come to conclusion that this is binding site for the 4-diphenylmethoxy-1-methylpiperidine derivatives and why not the same binding site as nifedipine considering it share roughly similar base structure as 4-diphenylmethoxy-1-methylpiperidine derivatives? Did the authors calculate the binding score of nifedipine with the binding site of 4-diphenylmethoxy-1-methylpiperidine derivatives? Generally, the docking score of hits can be compared with the known inhibitor in the same binding site. However, in this case, the authors compared the docking score of  hits with nifedipine in different binding sites. This is not a proper docking protocol to validate and interpret the docking score of the hits.

4) Figure 1A and 1B shows the docking poses of three diphenylmethoxypiperidine derivatives. However, from the figure, it looks like all the three molecules (red, green. Blue)  are the same as they have the same length of alkyl group chains on the piperidine ring.

5)     The interpretation of molecular interactions with respect to the docking score of diphenylmethoxypiperidine derivatives is missing. For example, the authors failed to explain why does the docking score of 4d molecule is less than 2b molecule despite it has an extra alkyl chain on the piperidine ring and it sits well in hydrophobic pocket formed by LEU 1237, ILU 213, LEU 217 and TYR 309 residues.

Author Response

chemistry-2073004

Molecular modeling and potential Ca2+ channel blocker activity of diphenylmethoxypiperidine derivatives

Response to Reviewer 1

(1)One of the serious concerns of this study is that the diphenylmethoxypiperidine derivatives don't show as effectiveness as nifedipine in functional studies and not even by 50% effective compared to nifedipine. This shows that these compounds are not effective to block the LTCC channels despite binding to one of the binding regions of LTCC channel with high affinity. The manuscript clearly missed the details of mechanism of action of how diphenylmethoxypiperidine derivatives binding in that region can trigger the blockage of the channel.

The docking study undertaken here was to detail a structure activity relationship of the diphenylmethoxypiperidine derivatives with respect to their experimentally determined inhibition of LTCC activity.  Docking scores are not directly translatable into IC50s but are useful to understand the structure activity relationship and guide synthesis in compound development and to optimize compounds for a predicted binding site. The reviewer is correct that affinity calculations for compounds that bind distinct/different sites are not comparable.

We believe that our results of docking analyses contribute to explaining our ex vivo functional results, however a detailed description of the mechanism(s) of channel blockade will require analyses outside the scope of this study.

(2)The following statement from lines 143-146 is incorrect and misleading as diphenylmethoxypiperidine derivatives don't interact at the same position as nifedipine and don’t block the channel as effectively as nifedipine.

“Our results showed that 4-diphenylmethoxy-1-methylpiperidine derivatives interact

with Ca2+ channel structure Cav1.1 in a manner similar to one of the clinically relevant LTCC blockers nifedipine.”

The intent of the statement was not to imply the 4-diphenylmethoxy-1-methylpiperidine derivatives are as effective as nifedipine.  We hypothesize that the observed ex vivo activity may be acting via a mechanism similar to nifedipine, as these compounds are predicted to bind a site adjacent to that observed for nifedipine in the cryo-EM structure (and recapitulated in our docking results).  The statement has been expanded and clarified.

This statement is now revised in Section 3.2 Docking Analysis, page 3, lane 120.

(3)How did the authors choose the binding site? Why did the authors consider the binding pocket adjacent to nifedipine for the 4-diphenylmethoxy-1-methylpiperidine derivatives and these details are missing. How did the authors come to conclusion that this is binding site for the 4-diphenylmethoxy-1-methylpiperidine derivatives and why not the same binding site as nifedipine considering it share roughly similar base structure as 4-diphenylmethoxy-1-methylpiperidine derivatives? Did the authors calculate the binding score of nifedipine with the binding site of 4-diphenylmethoxy-1-methylpiperidine derivatives? Generally, the docking score of hits can be compared with the known inhibitor in the same binding site. However, in this case, the authors compared the docking score of hits with nifedipine in different binding sites. This is not a proper docking protocol to validate and interpret the docking score of the hits.

The docking of the 4-diphenylmethoxy-1-methylpiperidine derivatives was done in an unbiased manner.  The complete protein was included in the docking grid(s) to allow the ligands to sample the entire protein surface and water accessible cavities. The docking scores presented in the manuscript are for the site having the lowest free energy of binding.  Nifedipine was removed from the structure file, included in the ligand library and re-docked.  The highest affinity site for nifedipine was that of the original cryo-EM structure supporting the accuracy of the docking method.  The docking scores of -8.2 to -8.4 kcal/mol are well within the known error of the docking algorithm and can be considered essentially the same for 2a, 2b and 4b and are not comparable to nifedipine.  The relative affinities are for the respective sites and shown for completeness.

This information is now included in Section 2. Materials & Methods, page 3, line 72.

(4)Figure 1A and 1B shows the docking poses of three diphenylmethoxypiperidine derivatives. However, from the figure, it looks like all the three molecules (red, green. Blue) are the same as they have the same length of alkyl group chains on the piperidine ring.

The figure has been modified to better show the different alkyl chains for each of the molecules, methyl for 2a, 2b and butyl for 4b.

New Figures 1A and 1B are now included.

(5)The interpretation of molecular interactions with respect to the docking score of diphenylmethoxypiperidine derivatives is missing. For example, the authors failed to explain why does the docking score of 4d molecule is less than 2b molecule despite it has an extra alkyl chain on the piperidine ring and it sits well in hydrophobic pocket formed by LEU 1237, ILU 213, LEU 217 and TYR 309 residues.

The variation in activity is attributed to differences in the halide moieties of the phenyl rings as the calculated free energies of binding for these compounds are essentially the same. To address concerns from Reviewer 2, a more exhaustive docking study has been completed and binding site residues updated and clarified in the manuscript. The methyl of 2a and 2b and the butyl group of 4a all sit in a pocket comprised of ILE 305, LEU 217 and PHE 218.  However, the longer butyl moiety extends further out and clashes with the epsilon oxygen of GLU 156.

Reviewer 2 Report

In this manuscript, Victor M. Pulgar and colleagues studied the role small molecules as potential Ca2+ blockers of VDCC ion channels using both experimental and in silico tools. The works shows an important finding the field, especially because the authors show that the compounds modulate the activity of Ca2+ channels using vascular tissues, the strength of this communication, were the authors have a very well stablish protocol and experience with these experiments (ref. 6 and 16)

The presented data take a step forward in the field, but the in silico results show discrepancies, and it is difficult to infer a structure-activity relationship with the results and how they are correlated to both Cav1.1 or Cav1.2 isoforms.

The manuscript cannot be published in the current version.

Concerns:

·       Please define “calcium” (Ca2+) (line 30) and correct Ca2+ to Ca2+ in the whole manuscript.

·       Please define α1c or make it consistent with line 40, α1c

·       To make it more clear to the readers, the reviewer would like to know how similar the Cav1.2 isoform is to the Cav1.1 isoform, since most of the known blockers interacts with the Cav1.2 isoform (as mentioned by the authors in line 44-47). Proteins sequence identity analysis might help to better understand how identical these active sites are.

·       The reported compounds (reference 6) published by the authors, shows very interesting results, and inhibit KCl-induced and noradrenaline-dependent contractions in mesenteric arteries ex vivo. Why the authors used molecular docking against the Cav1.1 isoform then?

·       Please check if the PDB has suitable structures for Cav1.2 for molecular docking (the reviewer checked the PDB) or if a homology model should be prepared for Cav1.2 based on the Cav1.1 cryo-EM structure.  

·       If the known blockers bind to the Cav1.2 isoform, why the authors predicted the binding against Cav1.1? Please compare the binding against both targets if there is no previous experimental evidence to suggest that the evaluated compounds bind to either Cav1.2 or Cav1.1 isoform. Is there any competitive binding ex vivo?

·       The 6JP5 structure, despite having a 2.9 Å resolution, it shows poor percentile ranks and metrics, including a clashcore and outliers in the red region compared to other structures with similar resolution, making the molecular docking predictions questionable.

·       It is not clear the dimensions of the grid box used for molecular docking, if they tested just the pore, and how the authors decided to pick the best poses — energies or clusters?  What does “…dock anywere” means? (line 72)

·       MM-GBSA rescoring of binding modes with Prime (Schrodinger’s suite) are required for a better ranking. Lines 89-95, including table 1. Molecular docking is an excellent tool to predict binding poses, but not to rank binding energies.

·        Why the authors are so sure about the compounds binding to Cav1.1 proteins? lines 143-145 and not to other type of Ca2+ channels?

·       Authors might make available the docking data if requested by readers. Line 181.

Author Response

chemistry-2073004

Molecular modeling and potential Ca2+ channel blocker activity of diphenylmethoxypiperidine derivatives

Response to Reviewer 2

(1)Please define “calcium” (Ca2+) (line 30) and correct Ca2+ to Ca2+ in the whole manuscript.

Corrected in the manuscript. See page 1, line 31.

(2)Please define α1c or make it consistent with line 40, α1c

Corrected in the manuscript. See page 1, line 40.

(3)To make it more clear to the readers, the reviewer would like to know how similar the Cav1.2 isoform is to the Cav1.1 isoform, since most of the known blockers interacts with the Cav1.2 isoform (as mentioned by the authors in line 44-47). Proteins sequence identity analysis might help to better understand how identical these active sites are.

We selected Cav 1.1 as it had the most complete structure available. Alignment of the available sequences for the isoforms Cav 1.2 (CACNA1C) and Cav 1.1 (CACNA1S) show that the residues predicted to interact with the three diphenylmethoxypiperidine derivatives are conserved between the isoforms. A sequence alignment is now presented for Cav1.1 isoforms (Figure 2A) and for Cav1.1 vs Cav1.2 (Figure 2B) showing the residues predicted to be involved in binding of the diphenylmethoxypiperidine derivatives studied.

See Figure 2, page 5, and section 3.3. Protein sequence alignment of LTCC alpha subunits, page 4, lane 141.

(4)The reported compounds (reference 6) published by the authors, shows very interesting results, and inhibit KCl-induced and noradrenaline-dependent contractions in mesenteric arteries ex vivo. Why the authors used molecular docking against the Cav1.1 isoform then?

The cryo-EM structure of cav 1.1 (pdb 6JP5) is the most complete structure and was solved with nifedipine, the control molecule for this study.  Since similar activity was observed experimentally in the ex vivo model, we investigated if these compounds would bind in the same site or perhaps an alternate site. They would presumably bind Cav 1.2 as well, we included now the information that the majority of the residues predicted to bind diphenylmethoxypiperidine derivatives are conserved in Cav1.2, but the structure for the transmembrane region is not available. 

(5)Please check if the PDB has suitable structures for Cav1.2 for molecular docking (the reviewer checked the PDB) or if a homology model should be prepared for Cav1.2 based on the Cav1.1 cryo-EM structure. 

The available Cav1.2 are not complete biological assemblies and do not contain the S6 domains where nifedipine binds and the adjacent regions where the diphenylmethoxypiperidine derivatives are predicted to bind.

(6)If the known blockers bind to the Cav1.2 isoform, why the authors predicted the binding against Cav1.1? Please compare the binding against both targets if there is no previous experimental evidence to suggest that the evaluated compounds bind to either Cav1.2 or Cav1.1 isoform. Is there any competitive binding ex vivo?

As addressed above, the transmembrane region where these compounds are predicted to bind (similar to nifedipine), to our knowledge is not currently available. We appreciate the reviewer’s suggestion; however we have not performed competitive binding with these diphenylmethoxypiperidine derivatives.

(7)The 6JP5 structure, despite having a 2.9 Å resolution, it shows poor percentile ranks and metrics, including a clashcore and outliers in the red region compared to other structures with similar resolution, making the molecular docking predictions questionable.

6JP5 is the only complete structure containing nifedipine that has been solved.  In our experience cryo EM structures are generally better for docking studies despite slightly lower resolution.

(8)It is not clear the dimensions of the grid box used for molecular docking, if they tested just the pore, and how the authors decided to pick the best poses — energies or clusters? What does “…dock anywere” means? (line 72)

The grid box(s) encompassed the entire protein allowing unbiased identification of the sites with the lowest predicted energies, thus allowing the ligands to sample the entire protein. The best poses were selected based on clusters within those sites.  The text was modified to clarify the methodology used for unbiased docking. This information is now included in Section 2. Materials & Methods, page 3, line 72.

(9)MM-GBSA rescoring of binding modes with Prime (Schrodinger’s suite) are required for a better ranking. Lines 89-95, including table 1. Molecular docking is an excellent tool to predict binding poses, but not to rank binding energies.

A more exhaustive docking exercise was undertaken to clarify the interacting residues in the transmembrane domain using Autodock Vina.  The predicted binding energies for 2a, 2b and 4b are well within the known error of the docking algorithm in this binding site and can be considered essentially the same, so a more detailed ranking was not performed.

(10)Why the authors are so sure about the compounds binding to Cav1.1 proteins? lines 143-145 and not to other type of Ca2+ channels?

We have expanded our original hypothesis to include the possibility for these compounds to bind Cav1.2 isoforms as well (see new Figure 2A and 2B). In addition, the possibility of interaction with others Calcium Channels is now commented on the manuscript’s discussion.

(11)Authors might make available the docking data if requested by readers. Line 181.

Docking results can be made available through Figshare under tereeves@wakehealth.edu. This information is now included in the Data Availability Statement section.

Round 2

Reviewer 1 Report

I reviewed the article and reviewers addressed my comments in the manuscript and response letter. I agree to accept the article in the current form. 

Reviewer 2 Report

1.     As addressed above, the transmembrane region where these compounds are predicted to bind (similar to nifedipine), to our knowledge is not currently available. We appreciate the reviewer’s suggestion; however we have not performed competitive binding with these diphenylmethoxypiperidine derivatives

It would be of great benefit to mention to the readers (probably in the conclusions) and make sure they understand that the structural study pretends to provide insights into potential binding interactions of the evaluated compounds with proteins involved on this calcium mechanisms as studied ex-vivo. With this, readers will understand that these binding interaction predictions are suggestions of potential binding sites and that more work is needed to structurally verify these predictions. Otherwise, the study will look, to different readers, like when this are absolute binding poses. Reviewer suggest to include this rational statement.

2.     6JP5 is the only complete structure containing nifedipine that has been solved. In our experience cryo EM structures are generally better for docking studies despite slightly lower resolution.

The reviewer disagrees. Despite what the authors indicate, no matter if the Cryo-EM structure pr X-ray crystal structure is available to a given protein, to determine which on is better. Generally, high resolution structures are better than low resolution structures, no matter the technique used to solve the structure — which is usually just one static pose of the global minimum of the protein and does not represent the dynamics of an ion channel in open or closed conformations.  It is of great benefit to have a ligand in complex in the cryo-EM structure, but molecular docking could also be used to predict the binding on other structures that do not have the ligand — for this reason, molecular dynamics simulations are powerful tools to make the system dynamic to further calculate energy binding, which was not carried out by the authors. The fact that the cryo-EM structure has a very high clashscore indicate that a molecular dynamic simulation should be carried out before the docking study, to relax the structure and conflictive amino acids after human data processing. So, not because cryo-EM provided insights about the complex means that is better for docking studies.

3.     A more exhaustive docking exercise was undertaken to clarify the interacting residues in the transmembrane domain using Autodock Vina. The predicted binding energies for 2a2b and 4b are well within the known error of the docking algorithm in this binding site and can be considered essentially the same, so a more detailed ranking was not performed.

The reviewer disagrees. Just because AutodockVina was not able to differentiate the energies between the compounds corroborates what the reviewer suggested to re-rank the docking poses using MM-GBSA. It is well know that docking energies are not a good option to establish structure-activity relationships, but it is the beest technique we have to predict a bindinig mode — please visit:

https://www.mdpi.com/1420-3049/23/5/1038

https://www.sciencedirect.com/science/article/abs/pii/S1476927117307521?via%3Dihub

The reviewer considers that the authors should indicate clearly in the manuscript (to strength the conclusions) that they do not pretend to stablish a structure-activity relationships by quantifying the docking energies, but they used these energies to select the best poses, reason why Autodock vina did not differentiate the energies. It would be of great benefit to reviewers to let them know that MM-GBSA studies are needed to better rank the docking poses once the binding site of these compounds could be corroborated. Please indicate that this work opens the possibility of designing novel potent derivatives based on 1a, 2b, and 4a, which the study really supports, beyond trying to propose a binding site on these channels. Please take advantage of it.

Author Response

please see attached file with a point-by-point response to Reviewer 2 comments.
